# Antero-Posterior Mandibular Excursion in Obstructive Sleep Apnea Patients Treated with Mandibular Advancement Device: A Retrospective Cohort Study

**DOI:** 10.3390/ijerph20043561

**Published:** 2023-02-17

**Authors:** Chiara Stipa, Serena Incerti-Parenti, Matteo Cameli, Daniela Rita Ippolito, Antonio Gracco, Giulio Alessandri-Bonetti

**Affiliations:** 1Unit of Orthodontics and Sleep Dentistry, Department of Biomedical and Neuromotor Sciences (DIBINEM), University of Bologna, Via San Vitale 59, 40125 Bologna, Italy; 2Department of Neurosciences, University of Padua, Via Giustiniani 3, 35128 Padua, Italy

**Keywords:** obstructive sleep apnea, mandibular advancement device, sleep dentistry

## Abstract

Since obstructive sleep apnea (OSA) management with a mandibular advancement device (MAD) is likely to be life-long, potential changes in mandibular movements during therapy should be investigated. The purpose of this study was to use a method that has already been shown to be reliable in order to determine whether the range of antero-posterior mandibular excursion, the procedure upon which MAD titration is based, varies between baseline (T0) and at least 1 year of treatment (T1). The distance between maximal voluntary protrusion and maximal voluntary retrusion determined using the millimetric scale of the George Gauge was retrospectively collected from the medical records of 59 OSA patients treated with the MAD and compared between T0 and T1. A regression analysis was performed to evaluate the influence of treatment time, MAD therapeutic advancement and the patient’s initial characteristics in excursion range variation. A statistically significant increase of 0.80 ± 1.52 mm (mean ± standard deviation, *p* < 0.001) was found for antero-posterior mandibular excursion. The longer the treatment time (*p* = 0.044) and the smaller the patient’s mandibular excursion at T0 (*p* = 0.002), the greater the increase was. These findings could be explained by a muscle–tendon unit adaptation to the forward mandibular repositioning induced by the MAD. During MAD therapy, patients can develop a wider range of antero-posterior mandibular excursion, especially those with a smaller initial excursion capacity.

## 1. Introduction

Obstructive sleep apnea (OSA) is a common chronic sleep-related breathing disorder characterized by the recurring upper-airway collapse during sleep, which causes snoring, cessation of breathing, intermittent hypoxia and sleep fragmentation [1]. The presence and the severity of OSA are currently defined by the apnea-hypopnea index (AHI), which is the number of episodes of complete or partial upper-airway obstruction per hour. Patients with OSA experience excessive sleepiness, neurocognitive deficiency and decreased quality of life. Increased risk of road and work accidents, cardiovascular morbidity and mortality are among the most severe complications [2,3]. Currently, continuous positive airway pressure (CPAP) constitutes the “gold standard” therapy for OSA. However, the high clinical effectiveness of CPAP is often limited by poor patient compliance [4].

The mandibular advancement device (MAD) is increasingly used for the treatment of patients with mild to moderate OSA or primary snoring and often represents an accepted therapy for patients with severe OSA who are unable or unwilling to tolerate CPAP [5,6]. MADs are contraindicated if there are (1) an insufficient number of teeth as anchorage, (2) active periodontal problems, (3) active temporomandibular disorders or (4) limited mandibular protrusive capacity (<6 mm) [7]. The function of the MAD is to mechanically keep the mandible in a forward position, thereby preventing the collapse of the upper airway during sleep. Despite the traditional thinking that the primary mechanism of action of the MAD is to increase the antero-posterior dimensions of the oropharynx, findings based on magnetic resonance imaging of the upper airway indicate that the MAD predominantly increases the volume of the velopharynx in its lateral dimensions [8]. Mandibular advancement also acts through the stabilization of the hyoid bone and the soft palate, the stretching of the tongue muscles and the prevention of the posterior rotation of the mandible in order to maintain a patent upper airway during sleep [8,9,10].

The MAD can be custom-made or prefabricated and, also, titratable (i.e., the mandible can be moved forward in increments over time) or non-titratable (i.e., the upper and lower dental arches are rigidly connected so that the mandible is kept in a single forward position). The American Academy of Sleep Medicine and the American Academy of Dental Sleep Medicine guidelines recommend the use of custom-made and titratable MADs, since they have been associated with better patient comfort and compliance with treatment [11,12,13]. The advancement of the MAD is generally expressed in terms of percentage of the range of antero-posterior mandibular excursion or in millimeters. The range of antero-posterior mandibular excursion is an important clinical characteristic upon which the construction bite of the MAD as well as the titration procedure are based. In fact, the MAD is fabricated at an initial mandibular position and then titrated to the therapeutic position. There is still considerable variation across studies on the recommended initial mandibular position, but the subsequent titration allows one to determine for each patient the minimal protrusion needed to achieve the highest reduction in the AHI while reducing patient discomfort and potential side effects [14,15,16]. Most of the previous studies analyzing the change in the range of antero-posterior mandibular excursion during treatment with the MAD did not indicate the method of excursion measurement, failed to indicate whether the maximal voluntary retrusion or the habitual bite position were used as posterior references and, also, used unreliable dental reference points [17,18,19,20,21,22].

While the anterior reference for the measurement was clearly defined as the maximum voluntary protrusion reached by the patient, there was no consensus about the posterior reference because both maximum voluntary retrusion and habitual bite position could be used for this purpose. These made it very difficult to compare the different studies with each other and, also, to develop well-defined recommendations for clinical practice. In a previous study [23], the ranges of antero-posterior mandibular excursion obtained using the two different posterior reference points were compared. The use of maximum voluntary retrusion provided a more reliable measurement compared with habitual bite position. This finding can be explained by the difficulties in reaching a repeatable habitual bite position during the measurement, when no teeth or cusps can make contact, whereas maximum retrusion is guided by the relationship between the condyles and fossae and therefore is not affected by this lack of occlusal contact [23]. Moreover, dental reference points (e.g., the distance from the margin of upper incisors to the margin of the lower incisors measured with a stainless steel ruler adopted by Fransson et al. [19]) can be considered unreliable for the measurements of the antero-posterior range of mandibular excursion. In fact, during long-term treatment with the MAD, the masticatory muscles attempt to restore the normal position of the mandible kept forward by the MAD, thus transmitting reciprocal forces to the teeth and determining a retroclination of the upper incisors and a proclination of the lower incisors, accompanied by a decrease in overjet and overbite [7].

Considering the protruded mandible position during sleep, an elongation of the muscle–tendon units and ligaments of the stomatognathic system can be expected during MAD wear. Several studies have shown a viscoelastic reaction of the muscle–tendon units to a mechanical stretching, which could determine their lengthening and could also improve the flexibility in human body joints over time [24,25,26,27,28,29,30,31]. Since the management of OSA is likely to be life-long, investigating the potential changes in mandibular movements during MAD therapy is an important issue.

Therefore, the first aim of this paper is to determine whether the range of antero-posterior mandibular excursion changes after at least 1 year of treatment with the MAD using the maximal voluntary protrusion and the maximal voluntary retrusion, respectively, as anterior and posterior references. A further objective is to evaluate the influence of treatment time, MAD therapeutic advancement and the patient’s initial characteristics on its variation.

## 2. Materials and Methods

### 2.1. Study Design and Data Collection

The study was designed as a retrospective cohort study. The protocol was approved by the Ethical Committee CE-AVEC of the AUSL of Bologna (number of approval: 44/2021/OSS/AUSLBO). Participants were retrieved from consecutive patients undergoing MAD treatment at the Unit of Orthodontics and Sleep Dentistry, Department of Biomedical and Neuromotor Sciences (DIBINEM), University of Bologna, Italy. Patients treated until 2020 were considered eligible for inclusion if they met the following requirements. The inclusion criteria were as follows: aged over 18 years, an established diagnosis of OSA based on polygraphic study according to the American Academy of Sleep Medicine criteria (i.e., AHI ≥ 5 with symptoms/sequelae or AHI ≥ 15 regardless of associated symptoms) [32], treatment with a custom-made titratable MAD that provided full coverage of the upper and lower dental arches (bi-block) and availability of the medical records of 2 measurements of the range of antero-posterior mandibular excursion before the beginning of therapy with the MAD (T0) and after at least 1 year of treatment (T1). The exclusion criteria were as follows: symptoms of temporomandibular disorders or myofascial pain at T0 and at T1, discontinuation of treatment with the MAD, use of the appliance for less than 5 nights per week or inability to wear the appliance for the entire night (assessed through a questionnaire).

Data surrounding the range of antero-posterior mandibular excursion at T0 and T1, treatment time (months between T0 and T1), therapeutic advancement of the MAD (percentage and millimeters of mandibular advancement at T0), age, sex, body mass index (BMI) and the AHI were collected. All data were obtained from manual medical record reviews by an operator (M.C.) that was blinded to the study aims.

All patients had undergone treatment with the MAD by an orthodontist qualified in dental sleep medicine (C.S. or M.C.). The measurement of the range of antero-posterior mandibular excursion was determined using the millimetric scale of the George Gauge (Great Lakes Orthodontics, Ltd., New York, NY, USA) according to the previously published methodology [23]: after being asked to sit straight in the dental chair, the patient was asked to move the mandible to the position of maximal voluntary protrusion and maximal voluntary retrusion which has been shown to provide a more reproducible measurement of the antero-posterior range of mandibular excursion (Figure 1a,b). The acquired data were used to obtain the construction bite of the MAD and represented the anatomical extreme movements to rely on when titrating the appliances. These measurements were routinely performed by one experienced operator qualified in dental sleep medicine (C.S. or M.C.) at both T0 and T1.

### 2.2. Statistical Analysis

Data were summarized using frequencies for nominal-level variables, while means and standard deviations (SD) were used for continuous data. The difference between the ranges of antero-posterior mandibular excursion before and after MAD therapy was assessed via a paired t-test. Linear regression analysis with a forward selection stepwise procedure was used to evaluate the association between the change in the range of antero-posterior mandibular excursion (dependent variable) and gender, age, BMI, the AHI, treatment time, therapeutic advancement of the device (expressed both in millimeters and in percentage of range of mandibular excursion) and the range of antero-posterior mandibular excursion at T0 (independent variables).

All statistical analyses were conducted by an operator (D.R.I.) using the Statistical Package for Social Sciences Software (IBM Corp., released 2012, IBM SPSS Statistics for Windows, Version 21.0., Armonk, NY, USA) at the 0.05 level of significance.

## 3. Results

The sample consisted of 59 OSA patients treated with the MAD. The main patient characteristics are presented in Table 1. The treatment time between T0 and T1 was 45.61 ± 27.86 (mean ± SD) months. The MAD therapeutic advancement was 9.28 ± 1.78 (mean ± SD) when expressed in millimeters and 72.47 ± 15.62% (mean ± SD) when expressed as a percentage of the range of mandibular excursion.

A statistically significant difference between the ranges of antero-posterior mandibular excursion at T0 (11.89 ± 1.95 mm) and T1 (12.69 ± 1.95 mm) was found, with a T0–T1 difference of 0.80 ± 1.52 mm (confidence interval (CI): 0.41–1.20 mm, *p* < 0.001).

The multiple stepwise linear regression model for the difference in the range of antero-posterior mandibular excursion (R^2^ = 0.210, *p* = 0.001) identified antero-posterior mandibular excursion at T0 (partial R^2^ = 0.151, *p* = 0.002) and treatment time (partial R^2^ = 0.060, *p* = 0.044) as the significant predictive variables (Table 2).

## 4. Discussion

In recent years, the concept of gradually increasing mandibular advancement until it achieves the most effective MAD therapeutic advancement personalized to each patient is emerging [16]. This approach should allow one to obtain the least possible amount of mandibular advancement while achieving the highest reduction in the AHI, thus optimizing the effectiveness of treatment while reducing patient discomfort and potential side effects both at the beginning of and during the therapy [14,15,16].

Although the titration procedure relies on the capacity of antero-posterior mandibular excursion, few studies have investigated the potential changes in the antero-posterior mandibular movement due to MAD therapy with conflicting outcomes [17,18,19,20,21,22]. No difference in mandibular protrusion capacity was found at the 2-year follow-up in a prospective cohort study investigating the craniomandibular status and function in patients with habitual snoring and OSA after nocturnal treatment with the MAD [17]. The study was conducted on 32 OSA patients treated with the MAD at a mandibular advancement varying between 50% and 70%; clinical examination included measurements of the range of movement of the mandible, the function of the temporomandibular joints, pain upon movement of the mandible and pain upon palpation of the temporomandibular joints and the masticatory muscles [17]. Similar results were obtained in a prospective randomized clinical trial comparing the effects of two different degrees of mandibular advancement (50% versus 75%) in 77 males with severe OSA after 6 months of MAD therapy in which a dentist examined the stomatognathic system, including the measurement of mandibular mobility, palpation of temporomandibular joints, masticatory muscles and pain upon mobility [18]. Conversely, a statistically significant increase emerged in two prospective cohort studies with a sample of 65 patients (44 with OSA and 21 with snoring treated at a 75% mandibular advancement) evaluated after 6 months and 2 years of MAD therapy as well as in a sample of 14 OSA patients at the 3-year follow-up (mean mandibular advancement of 77.2%) [19,20]. In the first study, the clinical examination included the assessment of mandibular mobility, tenderness to palpation of the temporomandibular joints and the masticatory muscles, function of the temporomandibular joints and pain upon mandibular movements. In the second study, a standardized clinical examination of the temporalis and masseter muscles, temporomandibular joints and jaw mobility was performed according to the Research Diagnostic Criteria for Temporomandibular Disorders (RDC/TMD). Similar results were obtained in a randomized clinical trial comparing the effect of 50% (38 patients) versus 75% mandibular advancement (36 patients), with a statistically significant increase in mandibular protrusion capacity being found in the latter group after 6 months of treatment [21]. In the latter study, the clinical examination included measurements of mandibular range of motion, overbite, overjet, palpation of the temporomandibular joints and masticatory muscles, registration of pain upon mobility and temporomandibular joint sounds.

These controversial results can be ascribed to different study designs and sample sizes. Moreover, a direct comparison with our findings would be inappropriate because all of the above mentioned studies analyzed a possible change in the capacity of mandibular protrusion at different time points of MAD therapy without taking into account the overall range of antero-posterior mandibular excursion. In the present study, we investigated the range of antero-posterior mandibular excursion using the maximum retrusion as a posterior reference because this method is recommended as being more reliable for MAD construction bite in order to identify the appropriate therapeutic position at the beginning of and during therapy [23]. Accordingly, the task force of the American Academy of Dental Sleep Medicine recommends that the posterior reference point be standardized to the most retruded position and that the posterior reference should be clearly indicated in future studies as well [33].

The majority of previous studies have not consistently indicated the method of mandibular protrusion measurement [17,18,19,20,21,22]. The study by Tegelberg et al. [21] reported that the mandibular range of motion was measured with a steel ruler to the nearest millimeter. The study by Fransson et al. [19] specified that a stainless steel ruler was used to obtain the distance from the margin of the upper incisors to the margin of the lower incisors. However, the reduction in overjet that occurs during long-term MAD therapy could influence this kind of measurement. Previous studies have demonstrated that during treatment with the MAD, there is a proclination of the lower incisors and a retroclination of the upper incisors resulting in an overjet reduction over time [34,35]. In the long term, the protrusion of the mandible induced by the MAD generates reciprocal forces on the soft tissues and the muscles which attempt to move the mandible backward to its normal position. These forces are transmitted to the teeth to which the MAD is anchored. A study conducted using cephalometric analysis after a mean period of 3.5 years of treatment with OAs showed a statistically significant retroclination of the upper incisors, and a proclination of the lower incisors accompanied by a decrease in overjet and overbite observed via three-dimensional dental cast analysis [34]. A more recent systematic review with a meta-regression analysis of 21 studies with a follow-up period varying between 2 and 11 years of treatment with the MAD confirms, as reported side effects, a reduction in overjet and overbite accompanied by a significant correlation between the duration of the therapy and the change in these parameters [35]. Therefore, the margin of the upper and lower incisors can be considered inappropriate as reference points for the measurement of the range of antero-posterior mandibular excursion. In the present study, the employment of the George Gauge, which reports the mandibular movement on a fixed millimetric scale, allows for the evaluation of the antero-posterior mandibular movement preventing systematic mistakes caused by dental movements over time. 

The main finding of the present study is the significant increase in the range of antero-posterior mandibular excursion after an average of 46 months of treatment with the MAD, with a mean T0-T1 difference of 0.80 ± 1.52 mm. This could be related to an adaptative change induced on the muscle–tendon units by the forward mandibular repositioning with the MAD, since it is demonstrated that stretching can increase the range of movement of the joints [36,37].

Two stretching proprioceptors located in muscle fibers and tendons (muscle spindles and Golgi tendon organs), respectively, relay information to the central nervous system and its reaction can lead muscles to adjust fibers and the sarcomere number to the functional length of the whole muscle [38]. Moreover, muscles can increase their protein synthesis and produce as many structural units as they need to obtain a correct synergy of myosin and actin bridges to perform proper activity [39,40,41]. This whole mechanism makes the muscles, held at a constant length, have decreased passive peak strength and viscosity. Therefore, the muscle–tendon units of the stomatognathic system could obtain further elasticity through the long-term mandibular protrusion induced by MAD use, which could account for the increase in the range of antero-posterior mandibular excursion found in the present study.

The same viscoelastic behavior can be ascribed to the temporomandibular joints’ ligaments as they are straightened for several hours during the night and this stretch is repeated every night during MAD therapy. They are subjected to a first elongation from the moment in which the device is applied, followed by another slower elongation for a second time during the whole night. When the MAD is removed, a part of the deformation is recovered, but another part will be regained only over time. If the force overcomes its elastic limit, the deformation could be enduring [29,30,31]. This further explains our result.

The biomechanical behavior of muscle–tendon units and ligaments could even account for the correlation found in the present study between treatment time and the increased range in the antero-posterior mandibular excursion. The longer the treatment time was, the greater the increase was in the mandibular excursion range. The continuous stimulation of muscle sarcomerogenesis and the deformation endured by ligaments and tendons caused by long-term MAD therapy could be the key to the interpretation of this result. 

Finally, we also found that the patient’s mandibular excursion range at T0 influenced the difference in the mandibular range of excursion between T0 and T1. The smaller the patient’s range of motion at T0 was, the greater the increase was in the mandibular antero-posterior movement. This result is crucial from a clinical point of view; in fact, a patient whose antero-posterior excursion is not really wide at T0 will have a larger gain in movement capacity than a patient whose mandibular range of motion is already wide at the baseline. It is therefore reasonable to assume that the currently accepted contraindication to MAD therapy in patients with a limited mandibular protrusive capacity (<6 mm) should be further investigated. As far as this result is concerned, we know that a larger advancement can determine a further chance to exert a positive effect on OSA, although no linear correlation is shown between the amount of advancement and the effectiveness of therapy [35,42,43]. 

Further prospective investigations should be conducted in order to improve the evidence level of our findings. All data obtained in this retrospective study were collected ex-post, so better standardization regarding the time in which clinical measurements are performed could be the objective for future investigation on this topic. It would also be worth evaluating other parameters such as AHI score and type of OSA in relation to the change in the antero-posterior range of mandibular excursion using a prospective study design. A further aim should be to better understand the biomechanical behavior of the MTUs and ligaments of the oro-pharyngeal region, as a change in their biomechanical properties could determine a stretch-induced and time-dependent adaptation to the forward mandibular repositioning induced by the MAD. 

## 5. Conclusions

The range of antero-posterior mandibular excursion increases during MAD therapy in a time-dependent manner. This measurement should be repeated during follow-up visits because the patient can develop a wider capacity of antero-posterior mandibular movement, even if the initial excursion capacity is limited.

## Figures and Tables

**Figure 1 ijerph-20-03561-f001:**
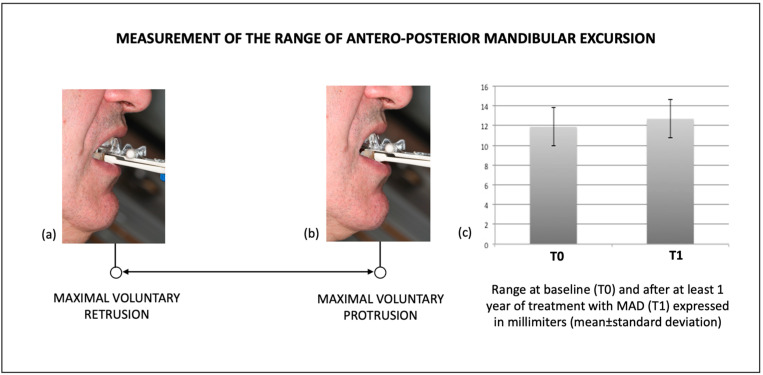
The range of antero-posterior mandibular excursion (retrospectively collected from medical records) had been routinely measured using the millimetric scale of the George Gauge (Great Lakes Orthodontics, Ltd., New York, NY, USA) as the distance (**a**) from the position of maximal voluntary retrusion (**b**) to the position of maximal voluntary protrusion.

**Table 1 ijerph-20-03561-t001:** Sample characteristics.

Males (Number (%))	44 (74.6%)
Age (years) (mean ± SD)	54.68 ± 9.07
BMI (kg/m^2^) (mean ± SD)	27.06 ± 2.70
AHI (events/h)	25.04 ± 15.52
Mild OSA (5 ≤ AHI < 15) (number (%))	18 (31%)
Moderate OSA (15 ≤ AHI < 30) (number (%))	18 (31%)
Severe OSA (AHI ≥ 30) (number (%))	23 (38%)

SD: standard deviation; BMI: body mass index; AHI: apnea hypopnea index; OSA: obstructive sleep apnea.

**Table 2 ijerph-20-03561-t002:** Linear regression model with the change in antero-posterior mandibular excursion as the dependent variable and baseline antero-posterior mandibular excursion and treatment time as the independent variables (stepwise regression).

	Estimate	Standard Error	T Value	*p* Value	95% CI (Lower Limit)	95% CI (Upper Limit)
Intercept	3.298	1.242	2.655	0.010	0.809	5.787
Range at T0	−0.262	0.095	−2.760	0.008	−0.452	−0.072
Treatment time	0.014	0.007	2.060	0.044	0.000	0.027

95% CI: 95% confidence interval for the estimate; range at T0: range of antero-posterior mandibular excursion at baseline.

## Data Availability

The data presented in this study are available upon request from the corresponding author.

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
