# Peer review of "Antero-Posterior Mandibular Excursion in Obstructive Sleep Apnea Patients Treated with Mandibular Advancement Device: A Retrospective Cohort Study"

_ijerph, 2023, doi:10.3390/ijerph20043561_

Round 1
Reviewer 1 Report
Hello,
Although the subject of the retrospective study is interesting, it has been studied before. What's new?
Abstract: The abstract presents the basic information in a concise manner.
Aim: The first goal has been analysed in many studies...what is the element of novelty? Perhaps it is worth evaluating other parameters (AHI score, type of OSA) in relation to the discoveries made in the case of the second goal. In general, the proposed aims were better for a prospective study.
Material and methods: Well written. Can you specify which is the most frequent MAD used in the studied group? Has the inter-examiner Kappa index been determined?
Regarding the statistical analysis, the sentence "All statistical analyzes were conducted by an operator (D.R.I.) that was blinded to the study aims" should be revised.
Results: Can you support the results from rows 125-127 with graphs or tables?
Discussions: Minor spelling changes.
Author Response
Response to Reviewer 1 Comments
Dear Reviewer 1,
we really appreciated your suggestions and found them of great interest in order to strengthen our work.
Hoping you will be satisfied with the revisions made to the paper, please find below a point-by-point reply to your comments.
Point 1: Comments and Suggestions for Authors
Hello, Although the subject of the retrospective study is interesting, it has been studied before. What's new?
Response 1: In the present study we investigated the range of antero-posterior mandibular excursion using a reliable method (the maximal voluntary protrusion was adopted as anterior reference and the maximal voluntary retrusion as posterior reference). The majority of previous studies did not indicate the method of mandibular excursion measurement and, also, failed to indicate whether the maximal voluntary retrusion or the habitual bite position was used as posterior reference, while a previous study (Ippolito et al. JOR 2020) showed the maximal voluntary retrusion rather than the habitual bite position as the most reliable posterior reference for mandibular advancement assessment. Among previous studies investigating the range of antero-posterior mandibular excursion in patients treated with MAD, only the study by Fransson et al. (AJO-DO 2004) indicated the measurement method (a stainless-steel ruler to obtain the distance from the margin of upper incisors to the margin of the lower incisors). However, the reduction in overjet that occurs during a long-term MAD therapy could influence this kind of measurement. These concepts are now emphasized at the end of the Introduction and, also, in the Discussion.
Point 2: Abstract: The abstract presents the basic information in a concise manner.
Response 2: Thank you very much.
Point 3: Aim: The first goal has been analysed in many studies...what is the element of novelty?
Response 3: The element of novelty has now been emphasized (please, see response 1).
Point 4: Perhaps it is worth evaluating other parameters (AHI score, type of OSA) in relation to the discoveries made in the case of the second goal. In general, the proposed aims were better for a prospective study.
Response 4: We agree with Reviewer 1 that a prospective design would be better in order to assess parameters such as AHI score and type of OSA in relation to the change in the antero-posterior range of mandibular excursion. This concept is now reported at the end of the discussion as recommendation for future research.
Point 5: Material and methods: Well written. Can you specify which is the most frequent MAD used in the studied group? Has the inter-examiner Kappa index been determined?
Response 5: As already reported in the Methods, the characteristics common to the devices used were “custom-made titratable MAD that provided full coverage of the upper and lower dental arches (bi-block)”. The most frequently used device was Somnomed.
Due to the retrospective design of the present study, no inter-examiner reliability index was determined. The measurement was routinely performed by one experienced operator qualified in Dental Sleep Medicine (C.S. or M.C.), as now explicitly reported in the Methods.
Point 6 Regarding the statistical analysis, the sentence "All statistical analyzes were conducted by an operator (D.R.I.) that was blinded to the study aims" should be revised.
Response 6: The sentence has been rephrased according to the given suggestion.
Point 7: Results: Can you support the results from rows 125-127 with graphs or tables?
Response 7: According to the given suggestion, a graph has now been added in Figure 1.
Point 8: Discussions: Minor spelling changes.
Response 8: Language has now been checked throughout the manuscript by an English expert.

Reviewer 2 Report
It is a remarkable manuscript on a very interesting subject. Thanks to the authors. I think that making the following minor corrections will be more valuable in terms of the presentation of the study.
Abstract: An explanatory sentence should be added to the material method part of this section. P values should also be mentioned while writing the findings.
Introduction: Well written.
Materials and methods: How were the measurements of the two orthodontists mentioned in lines 94-96 calibrated with each other? Has a preliminary study been done for the calibration of the measurements? Please specify clearly.
Results: Well written.
Discussion: Is there any literature supporting the data in the sentence “The longer the treatment time, the greater was the increase in the mandibular excursion range.” mentioned on lines 205-206? If there is, add it and give details and compare.
References: References 13, 14, 16, 17, 18, 19, 21, 22, 24, 25, 26, 35, 36, 37, 38 and 39 are very old. Please rewrite what is possible with new references.
Author Response
Response to Reviewer 2 Comments
Dear Reviewer 2,
we really appreciated your suggestions and found them of great interest in order to strengthen our work.
Hoping you will be satisfied with the revisions made to the paper, please find below a point-by-point reply to your comments.
Point 1: It is a remarkable manuscript on a very interesting subject. Thanks to the authors. I think that making the following minor corrections will be more valuable in terms of the presentation of the study.
Response 1: Thank you very much.
Point 2: Abstract: An explanatory sentence should be added to the material method part of this section. P values should also be mentioned while writing the findings.
Response 2: According to the given suggestion, the methods are now described in more details. P values have also been added.
Point 3: Introduction: Well written.
Response 3: Thank you very much.
Point 4: Materials and methods: How were the measurements of the two orthodontists mentioned in lines 94-96 calibrated with each other? Has a preliminary study been done for the calibration of the measurements? Please specify clearly.
Due to the retrospective design of the present study, the two orthodontists were not calibrated each other. However, as now explicitly reported in the Methods, the measurement was routinely performed by one experienced operator qualified in Dental Sleep Medicine (C.S. or M.C.). Moreover, a reliable measurement method was chosen according to the results of a previous study in which the use of maximal voluntary retrusion provided a more reliable measure compared to the use of habitual bite position as a posterior reference (Ippolito et al. JOR 2020).
Point 5: Results: Well written.
Response 5: Thank you very much.
Point 6: Discussion: Is there any literature supporting the data in the sentence “The longer the treatment time, the greater was the increase in the mandibular excursion range.” mentioned on lines 205-206? If there is, add it and give details and compare.
Response 6: This sentence was based upon the findings of the present study. Unfortunately, no direct comparison ca be made with previous studies.
Point 7: References: References 13, 14, 16, 17, 18, 19, 21, 22, 24, 25, 26, 35, 36, 37, 38 and 39 are very old. Please rewrite what is possible with new references.
Response 7: Thank you for the given suggestion. Much effort has been made to cite more recent articles.

Reviewer 3 Report
Dear authors,
congratulations on your work. I have a few comments-
the introduction is very short -please describe the effects of OSA and how your described method is different from other available methods. how mandibular advancement is beneficial.
Aim of the study should be re-written -"Therefore, the first aim of this paper is to determine whether this variable 60 changes after at least 1 year of treatment with MAD. The second aim is to evaluate the 61 influence of treatment time, MAD therapeutic advancement and patient’s initial charac- 62 teristics on its variation."
there should be one aim then you can have objectives .....
in methodology explain the methods separately and clearly, and describe sample size selection and its basis.
limitations of your study ?
and the significance of the study should be highlighted.
change references many ref. are too old..add new references
please get your work checked by an English expert...
Author Response
Response to Reviewer 3 Comments
Dear Reviewer 3,
we really appreciated your suggestions and found them of great interest in order to strengthen our work.
Hoping you will be satisfied with the revisions made to the paper, please find below a point-by-point reply to your comments.
Point 1: Dear authors, congratulations on your work. I have a few comments- the introduction is very short -please describe the effects of OSA and how your described method is different from other available methods. how mandibular advancement is beneficial.
Response 1: Thank you very much. The effects of OSA, the difference between our measurement method and previous methods as well as the advantages of mandibular advancement have now been described in the Introduction.
Point 2: Aim of the study should be re-written -"Therefore, the first aim of this paper is to determine whether this variable changes after at least 1 year of treatment with MAD. The second aim is to evaluate the influence of treatment time, MAD therapeutic advancement and patient’s initial characteristics on its variation." there should be one aim then you can have objectives .....
Response 2: The aim of the study has been rephrased according to the given suggestion.
Point 3: in methodology explain the methods separately and clearly, and describe sample size selection and its basis.
Response 3: Consecutive patients treated with MAD until 2020 were considered eligible for inclusion in the study if they met the inclusion and exclusion criteria. This is now explicilty reported in the methods.
Point 4: limitations of your study ?
Response 4: Limitations are now reported at the end of the discussion.
Point 5: and the significance of the study should be highlighted.
Response 5: The significance of the study has now been highlighted in the Introduction and in the Discussion.
Point 6: change references many ref. are too old..add new references
Response 6: Thank you for the given suggestion. Much effort has been made to cite more recent articles.
Point 7: please get your work checked by an English expert...
Response 7: Language has now been checked throughout the manuscript by an English expert.

Round 2
Reviewer 3 Report
dear authors, congratulations on your work...many suggestions are implemented.
just correct the last paragraph of your introduction ..".Therefore, the first aim of this paper is to determine whether the range of antero-posterior mandibular excursion changes after at least 1 year of treatment with MAD using the maximal voluntary protrusion and the maximal voluntary retrusion, respectively, as anterior and posterior reference. A further objective is to evaluate the influence of treatment time, MAD therapeutic advancement and patient’s initial characteristics on its variation."
there is no first or last aim...AIM Is one only...and this study is already completed so ...it will be better to use past tense here .....WAS in place of is...
english needs to be corrected